# Neural étendue expander for ultra-wide-angle high-fidelity holographic display

Ethan Tseng [1], Grace Kuo[2], Seung-Hwan Baek[1,3], Nathan Matsuda[2], Andrew Maimone[2], Florian Schiffers[2], Praneeth Chakravarthula[1], Qiang Fu [4], Wolfgang Heidrich [4], Douglas Lanman[2] & Felix Heide [1]✉

Holographic displays can generate light fields by dynamically modulating the wavefront of a coherent beam of light using a spatial light modulator, promising rich virtual and augmented reality applications. However, the limited spatial resolution of existing dynamic spatial light modulators imposes a tight bound on the diffraction angle. As a result, modern holographic displays possess low étendue, which is the product of the display area and the maximum solid angle of diffracted light. The low étendue forces a sacrifice of either the field-of-view (FOV) or the display size. In this work, we lift this limitation by presenting neural étendue expanders. This new breed of optical elements, which is learned from a natural image dataset, enables higher diffraction angles for ultra-wide FOV while maintaining both a compact form factor and the fidelity of displayed contents to human viewers. With neural étendue expanders, we experimentally achieve 64 × étendue expansion of natural images in full color, expanding the FOV by an order of magnitude horizontally and vertically, with high-fidelity reconstruction quality (measured in PSNR) over 29 dB on retinal-resolution images.

Holography is the science of creating vivid scenery through carefully crafted interference patterns. This discipline has applications across domains, especially in virtual and augmented reality devices[1,2]. While static holograms can be generated with a suitable recording medium, modern holographic displays typically employ spatial light modulators (SLM) that dynamically modulate the wavefront of a coherent beam[3,4]. However, despite being the workhorse of holography, SLMs suffer from small diffraction angles caused by limitations of modern liquid crystal on silicon (LCoS) technology. Achieving dynamic control with LCoS induces several engineering challenges (e.g., display bandwidth, pixel cross-talk, power consumption), which have imposed a practical lower bound on the pixel pitch[5]. Consequently, the étendue of holographic displays, which is the product of the viewing angle and the display size for table-top displays and is the product of the field-of-view (FOV) and the eyebox size for near-eye displays, is fundamentally limited when maintaining a compact SLM size[6]. Holographic displays

have to trade off FOV for display size, or vice versa, although both are critical for most display applications. For immersive virtual/augmented reality (VR/AR) devices, an FOV of at least 120° and an eyebox size greater than 10 × 10 mm² is desired[7], where the eyebox or display size is defined as the region within which the eye must reside within to view a hologram. To reach the étendue needed for these specifications requires over one billion SLM pixels, which is two orders of magnitude more than what today's LCoS technology achieves[7]. Manufacturing such a display and dynamically controlling it is beyond modern fabrication and computational capabilities.

Several methods have been proposed to circumvent this problem, including dynamic feedback in the form of eye tracking[8], spatial integration with multiple SLMs[9], and temporal integration with laser arrays[10] or fast switching digital micromirror devices[11,12]. However, these approaches require additional dynamic components resulting in high complexity, large form factors, precise timing constraints, and

[1]Department of Computer Science, Princeton University, Princeton, NJ, USA. [2]Reality Labs Research, Meta, Redmond, WA, USA. [3]Department of Computer Science and Engineering, Pohang University of Science and Technology (POSTECH), Pohang, Republic of Korea. [4]Visual Computing Center, King Abdullah University of Science and Technology (KAUST), Thuwal, Saudi Arabia. ✉e-mail: fheide@princeton.edu

incur additional power consumption. Timing is especially critical for eye tracking solutions as poor latency can result in motion sickness[13]. Rewritable photopolymers are promising alternatives to SLMs for holographic displays, however, they are limited by low refresh rates[14,15]. Microelectromechanical systems have low pixel counts and bit depth[16]. Researchers have also proposed to make trade-offs within the limited étendue of existing holographic displays, such as trading spatial resolution for depth resolution[3] and optimizing for coherent versus incoherent interference[17], but these methods do not change the total étendue of the display.

Instead, researchers have explored expanding the display étendue by employing optical elements with randomized scattering properties in front of an SLM[7,18–22]. The static nature of these elements facilitates fabrication of pixel areas at the micron-scale, an order of magnitude lower than for a dynamic element such as the SLM, thus resulting in an enlarged diffraction angle[23]. However, existing elements of this type exhibit randomized scattering that is agnostic to the optical setup and the images to be displayed. As modern SLMs have limited degrees of freedom for wavefront shaping, the random modulation delivered by these scattering elements results in low-fidelity étendue expanded holograms. Relaxing high frequencies beyond the human retinal resolution improves the fidelity, however the displayed holograms still suffer from low reconstruction fidelity[7]. Furthermore, extensive calibration is necessary in the case where the scattering properties are unknown[22,24–27]. Recent work explores using lenses and lenslet arrays[28] to expand the field-of-view but this approach forces the eye pupil's position to match that of the lenses or lenslets, thus shrinking the effective eyebox size, see Supplementary Note 5 for comparison experiments. Another line of work investigates tilting cascades[29], however, this system has a large physical footprint consisting of several 4F relays. Specifically, this system requires one 4F relay per $2\times$ factor of étendue expansion, thus three 4F relays are needed to achieve $8\times$ expansion along one axis, spanning roughly a meter in length. Moreover, the approach requires two parallel cascades to expand the étendue in both the horizontal and vertical directions, further increasing the physical form factor.

In this work, we lift these limitations with neural étendue expanders, a new breed of static optical elements that have been optimized for étendue expansion and accurate reproduction of natural images when combined with an SLM (Fig. 1). These optical elements inherit the aforementioned benefits of scattering elements. However, unlike existing random scattering masks, neural étendue expanders are jointly learned together with the SLM pattern across a natural image dataset. We devise a differentiable holographic image formation model that enables learning via first-order stochastic optimization. The resulting learned wavefront modulation pushes reconstruction noise outside of the perceivable frequency bands of human visual systems while retaining perceptually critical frequency bands of natural images. We provide analysis that validates that the learned optical elements possess these properties. In simulation, we demonstrate étendue-expanded holograms at $64\times$ étendue expansion factor with perceptual display quality over 29 dB peak signal-to-noise ratio (PSNR), more than one order of magnitude in reduced error over the existing methods. Furthermore, the expanders also facilitate high-fidelity étendue expanded 3D color holograms of natural scenes, see Supplementary Information, and the elements are also robust to varying pupil positions within the eyebox. We experimentally validate the method with a 1K-pixel SLM to expand the FOV by $8\times$ in each direction.

## Results

### Neural étendue expansion
Holographic displays modulate the wavefront of a coherent light beam using an SLM to form an image at a target location. The étendue of a holographic display[30] is then defined as the product of the SLM display area $A$ and the solid angle of the diffracted light as

$$G_s = 4A\sin^2\theta_s, \tag{1}$$

where $\theta_s = \sin^{-1}\frac{\lambda}{2\Delta_s}$ is the maximum diffraction angle of the SLM, $\lambda$ is the wavelength of light, and $\Delta_s$ is the SLM pixel pitch. Most SLMs have large $\Delta_s$ resulting in small $\theta_s$ as shown in Fig. 1a. We enlarge the display étendue by placing a neural étendue expander in front of the SLM as a static optical element with pixel pitch $\Delta_n < \Delta_s$, see Fig. 1b. The smaller pixel pitch $\Delta_n$ increases the maximum diffraction angle $\theta_n$, resulting in an expanded étendue $G_n = 4A\sin^2\theta_n$.

To generate étendue-expanded high-fidelity holograms, we propose a computational inverse-design method that learns the wavefront modulation of the neural étendue expander by treating it as a layer of trainable neurons that are taught to minimize a loss placed on the formed holographic image, see Fig. 1c. Specifically, we model the holographic image formation in a fully differentiable manner following Fourier optics. We relate the displayed holographic image $I$ to the wavefront modulation of the neural étendue expander $\mathcal{E}$ as

$$I = |\mathcal{F}(\mathcal{E} \odot U(\mathcal{S}))|^2, \tag{2}$$

where $\mathcal{F}$ is the 2D Fourier transform, $\mathcal{S}$ is the SLM modulation, $U(\cdot)$ is zeroth-order upsampling operator from the low-resolution SLM to the high-resolution neural étendue expander, and $\odot$ is the Hadamard product.

The differentiability of Eq. (2) with respect to the modulation variables $\mathcal{E}$ and $\mathcal{S}$ allows us to learn the optimal wavefront modulation of the neural étendue expander $\mathcal{E}$ by jointly optimizing the static neural étendue expander in conjunction with the dynamic SLM modulation patterns $\mathcal{S}$. That is, for a given image, we optimize the optimal SLM pattern similar to conventional computer-generated holography[31–33], however, we also simultaneously optimize the neural étendue expander. The SLM and the neural étendue expander cooperate to generate an étendue-expanded high-fidelity hologram. We formulate this joint optimization as

$$\underset{\mathcal{E},\mathcal{S}_{\{1,\dots,K\}}}{\text{minimize}} \sum_{k=1}^{K} \left\| \left( |\mathcal{F}(\mathcal{E} \odot U(\mathcal{S}_k))|^2 - T_k \right) * f \right\|_2^2, \tag{3}$$

where $\mathcal{S}_k$ is the SLM wavefront modulation for the $k$-th target image $T_k$ in a natural-image dataset with $K$ training samples, $*$ is the convolution operator, and $f$ is the low-pass Butterworth filter for approximating the viewer's retinal resolution as a frequency-cutoff function[7] as

$$f = \mathcal{F}^{-1}\left( \left( 1 + \left( \frac{\|w\|^2}{c^2} \right)^5 \right)^{-1} \right), \tag{4}$$

where $\mathcal{F}^{-1}$ is the inverse 2D Fourier transform, $w$ is the spatial frequency, and $c$ is the cutoff frequency. We set $c$ to be $\frac{\Delta_n N}{\sqrt{\pi}}$ where $N$ is the SLM pixel count. This allows us to utilize the native bandwidth of the SLM for only the lower frequency features that the human viewer will perceive. We can then adjust the FOV and the eyebox size so that an angular resolution of at least 60 pixels/degree is achieved, see Supplementary Note 3 for details.

The optimization objective in Eq. (3) jointly optimizes a single static element $\mathcal{E}$ and a set of SLM patterns $\mathcal{S}_{\{1,\dots,K\}}$ so that the set of generated holograms matches the target set of natural images $T_{\{1,\dots,K\}}$. This objective function is fully differentiable with respect to the wavefront modulations of the SLM and the neural étendue expander. As such, we can solve this optimization problem by training the neural étendue expander and the SLM states akin to a shallow neural network by using stochastic gradient solvers[34]. Our computational design approach is data-driven and requires a dataset of natural images. We

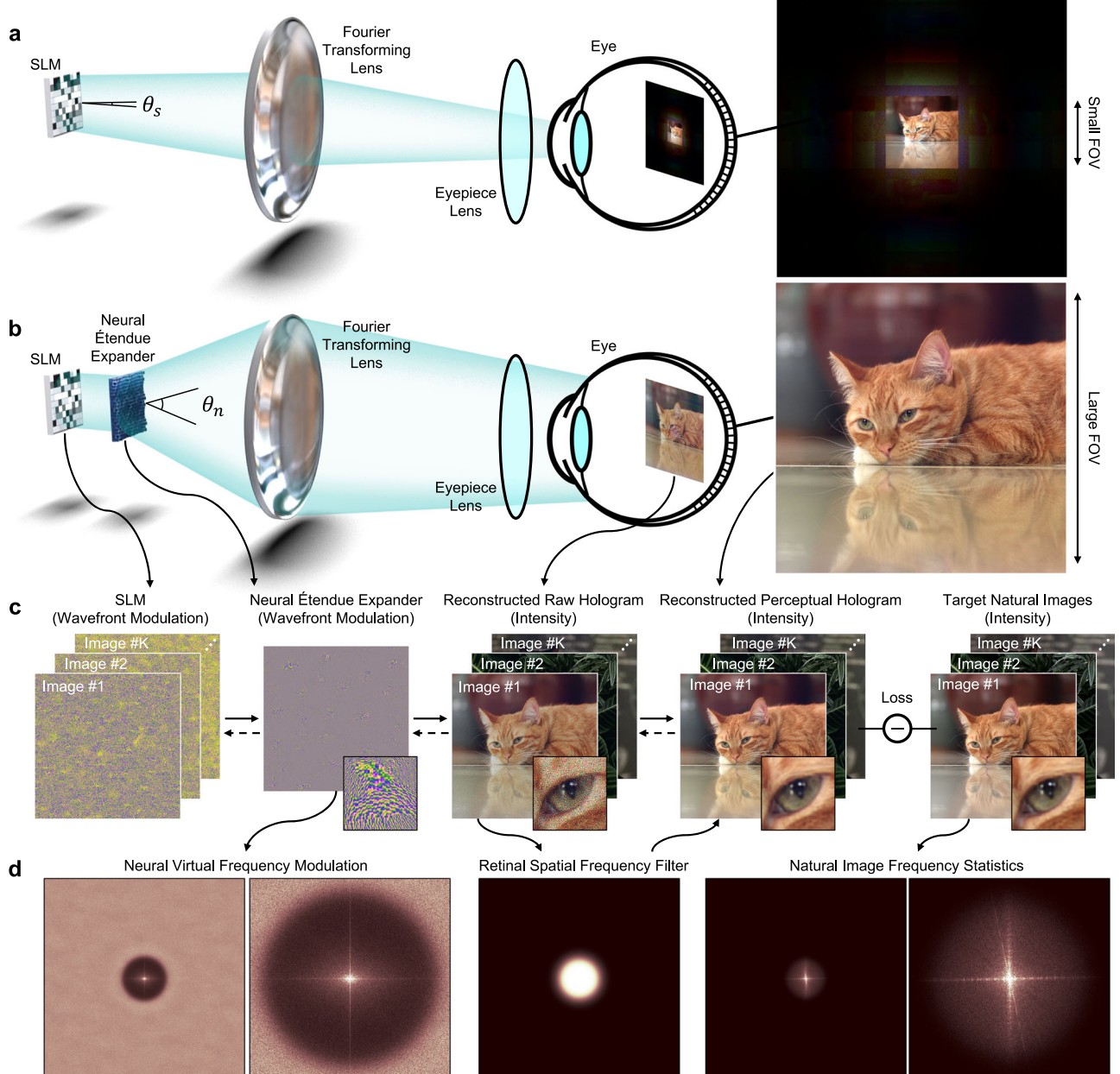

**Fig. 1 | Neural étendue expansion for ultra-wide angle, high-fidelity holograms.**
**a** Conventional holographic displays suffer from low étendue, resulting in either small FOV or eyebox size. Here, we illustrate a small FOV Fourier hologram corresponding to diffraction angle $\theta_s$. **b** Introducing a neural étendue expander into the display facilitates ultra-wide angle holograms. Here, we illustrate the increased FOV Fourier hologram corresponding to a larger diffraction angle $\theta_n$. **c** We design the neural étendue expanders via an end-to-end optimization algorithm that considers the SLM wavefront modulation and the human viewer's perceptual response. One SLM pattern is optimized for each training sample, while the neural étendue expander learns a general structure that facilitates hologram generation of any natural image. **d** The learned neural étendue expander preserves the major frequency bands of natural images within the frequency cutoff determined by the resolution of the human retina.

used 105 high-resolution training images of natural scenes. For testing, we used 20 natural images. We use grayscale images when designing neural étendue expanders for a monochromatic display, while the original RGB images are used when designing for a trichromatic display. The training procedure does not utilize any temporal multiplexing and only optimizes for the maximal reconstruction quality that can be achieved with a single SLM frame and a single expander.

## Neural étendue expanded holographic display
We validate neural étendue expansion experimentally with a holographic display prototype. See Fig. 2a for a schematic of the hardware prototype and Supplementary Notes 9 and 10 for further details on the experimental setup. We fabricate neural étendue expanders with a pitch of $2\,\mu$m with resin stamping, see Supplementary Note 8 for fabrication details. The fabricated expanders are then placed at the conjugate plane of the SLM to establish pixel-wise correspondence between the SLM and the expander. A DC block is further employed to filter out the undiffracted light from the SLM. To assess the proposed elements, we also compare to fabricated binary random expanders[7] designed for 660 nm. Microscope images of both expanders are shown in Fig. 2b. We acquire holograms corresponding to conventional non-étendue expanded holography[1], 64 × étendue expanded full

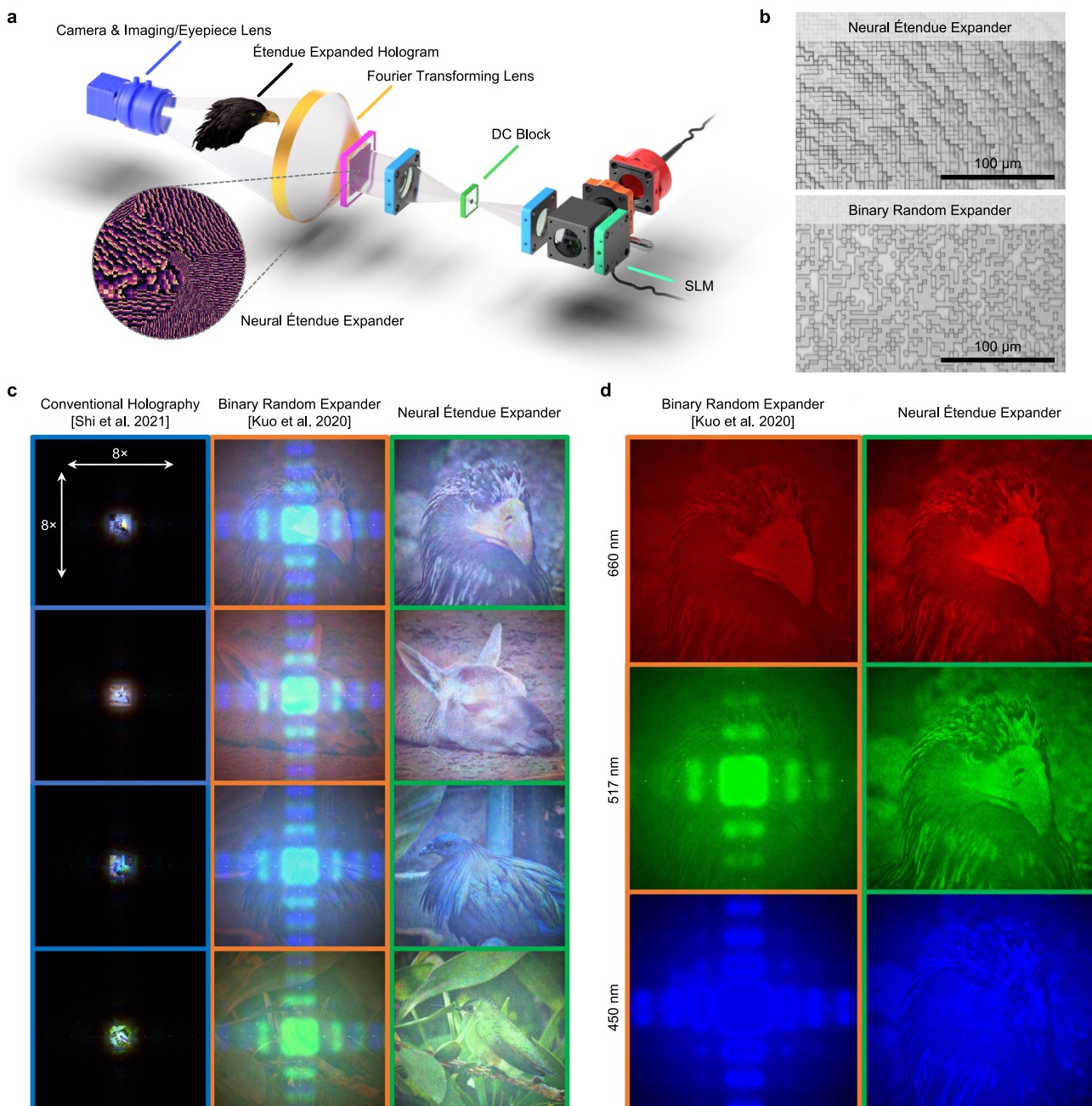

**Fig. 2 | Experimental demonstration of neural étendue expansion. a** Schematic of holographic display prototype with the neural étendue expander inserted at the conjugate plane of the SLM. This prototype generates étendue expanded Fourier holograms with a Fourier transforming lens and an eyepiece/imaging lens. **b** Microscope images of a fabricated neural étendue expander and a binary random expander[7]. **c** Captures of holograms generated with the display prototype. The small dark circle in the center of the pictures corresponds to the DC block. Left: Non-étendue expanded holograms produced with conventional holography[1]. These holograms have extremely low FOV. Middle: 64 × étendue expanded

holograms produced with the binary random expander show low contrast and chromatic artifacts. Right: 64 × étendue expanded holograms produced with the neural étendue expander show high fidelity. Temporal averaging with 20 frames was applied for the experimental results shown in this figure. See the Supplementary Information for raw single-frame experimental results. **d** Decomposition of 64 × étendue expanded holograms into constituent colors (450 nm, 517 nm, 660 nm). We observe improved hologram contrast and less speckle with neural étendue expansion, even at the wavelength of 660 nm which was used to design the binary random expander.

color holograms produced with the binary random expanders, and 64 × étendue expanded full color holograms produced with the neural étendue expanders. The illumination wavelengths are 450 nm, 520 nm, and 660 nm. We report captures in Fig. 2c and provide additional measurements in Supplementary Video 2 and in Supplementary Note 1. The captured holograms are tone-mapped for visualization. For fair comparison we applied the same tone-mapping scheme to all holograms, see Supplementary Note 1 for details.

The experimental findings on the display prototype verify that conventional non-étendue expanded holography can produce high-fidelity content but at the cost of a small FOV. Increasing the étendue via a binary random expander will increase the FOV but at the cost of low image fidelity, even at the design wavelength of 660 nm, and chromatic artifacts. The étendue expanded holograms produced with the neural étendue expanders are the only holograms that showcase both ultra-wide-FOV and high-fidelity. The captured holograms

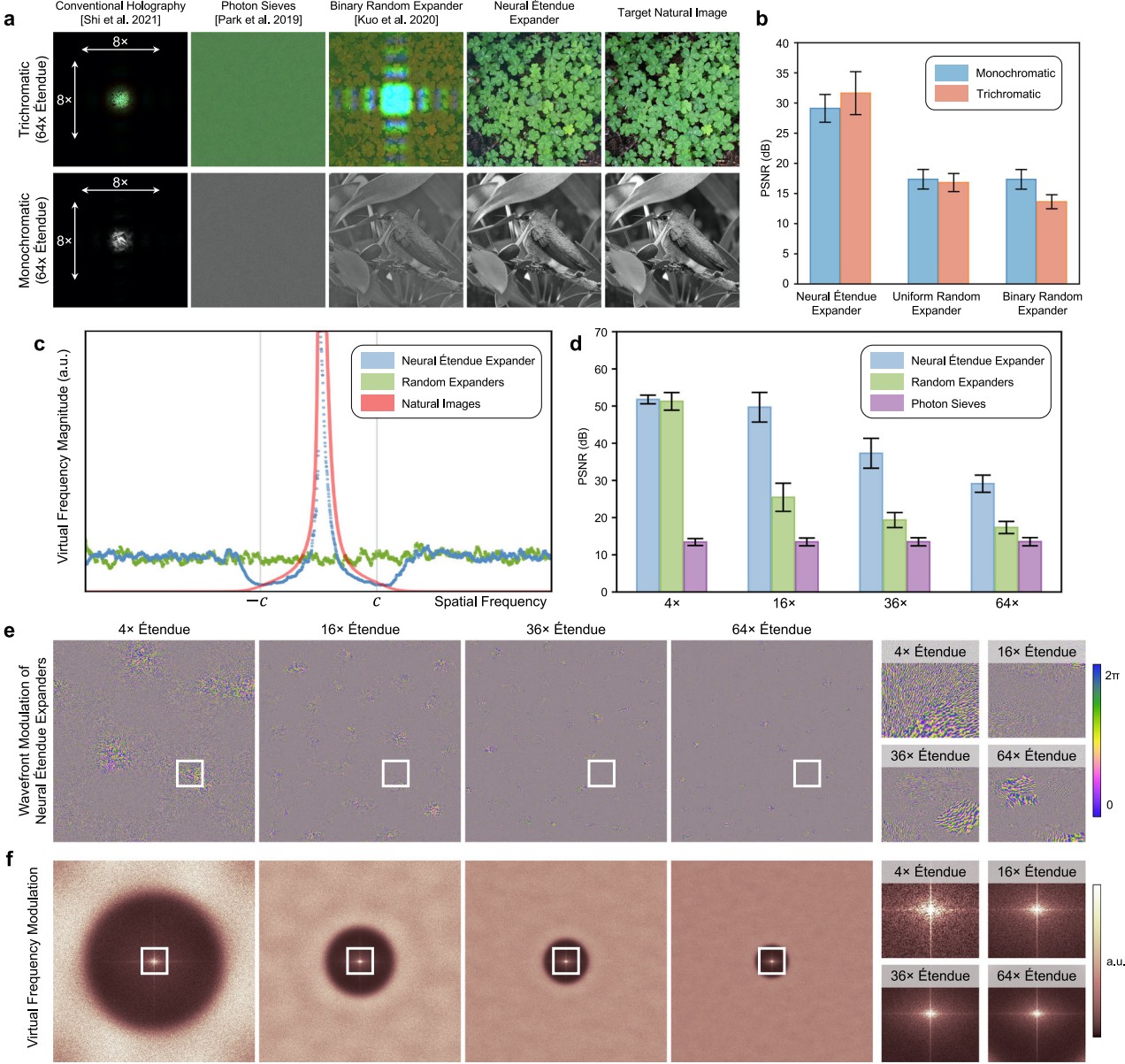

**Fig. 3 | Étendue expander characterization. a** The 64 × étendue expanded holograms generated with neural étendue expanders have the highest fidelity with respect to the target natural image, for both the trichromatic and monochromatic cases. In comparison, the holograms generated with binary random expanders[7] or photon sieves[21] show lower contrast and more speckle noise. Photon sieves could generate étendue expanded holograms of sparse points but not of natural scenes. A low étendue hologram generated with conventional holography[1] and no expander is included for comparison. **b** Quantitative performance comparison of 64 × étendue expanded holograms when using neural, uniform random, and binary random expanders. The metrics are evaluated over an unseen test set. **c** Virtual frequency modulation cross-section. Neural étendue expanders push reconstruction artifacts outside of the perceivable frequency bands of human vision while producing a

natural image frequency spectrum within the passband as predicted by Eq. (6). In contrast, the both uniform and binary random expanders exhibit a flat spectrum which reduces the reconstruction quality within the passband. The cutoff frequency is indicated by *c*. **d** Quantitative reconstruction quality of neural étendue expansion, random expansion, and photon sieves[21] for increasing étendue expansion factors for the monochromatic case. Uniform and binary random expansion both achieve the same performance for a single wavelength. **e** Visualization of the learned expanders for increasing étendue expansion factors. We observe that the learned modulation structures contain both high and low frequency components. **f** Visualization of the corresponding virtual frequency modulation for each expander. All error bars correspond to one standard deviation. a.u. stands for arbitrary units.

demonstrate high contrast and are free from chromatic aberrations. Fig. 2d reports the étendue expanded hologram produced with both expanders at each color wavelength. Since the binary random expander is, by design, only tailored to a single wavelength, in this case 660 nm, the étendue expanded holograms that are generated with it exhibit severe chromatic artifacts. In contrast, holograms generated with neural étendue expansion show consistent high-fidelity performance at all illumination wavelengths. Notably, even at the wavelength of 660 nm that was used to design the binary random expander[7] the

hologram fidelity is higher for the holograms generated with neural étendue expansion, see Fig. 2c. For comparisons against a uniform random expander, where the phase profile is uniformly randomly selected from within [0, 2π], see Fig. 3 and Supplementary Note 2.

While our experimental prototype was built for a HOLOEYE-PLUTO with 1K-pixel resolution, corresponding to a 1 mm eyebox with 77.4° horizontal and vertical FOV, the improvement in hologram fidelity persists across resolutions. Irrespective of the resolution of the SLM, performing 4 ×, 16 ×, or 64 × étendue expansion with neural

étendue expanders results in a similar margin of improvement over uniform and binary random expanders. This is because the improvement in fidelity depends only on the étendue expansion factor. We validate this by simulating an 8K-pixel SLM with 64 × étendue expansion and we verify that the improvement in fidelity is maintained. See Supplementary Note 6 for results and further details. Thus, neural étendue expansion enables high fidelity expansion for 64 × étendue expansion for 8K-pixel SLMs[35], providing étendue to cover 85% of the human stereo FOV[36] with a 18.5 mm eyebox size, see Supplementary Note 3 for details.

## Characterization of étendue expansion

Next, we analyze the expansion of étendue achieved with the proposed technique. To this end, suppose we want to generate the étendue-expanded hologram of only a single scene. Then, the optimal complex wavefront modulation for the neural étendue expander would be the inverse Fourier transform of the target scene, and, as such, we do not require any additional modulation on the SLM. The SLM therefore can be set to zero-phase modulation. If we generalize this single-image case to diverse natural images, the neural étendue expander is expected to preserve the common frequency statistics of natural images, while the SLM fills in the image-specific residual frequencies to generate a specific target image. In contrast, existing random scatters used for étendue expansion do not consider any natural-image statistics[7,18,20,21].

To assess whether the optimized neural étendue expander $\mathcal{E}$, shown in Fig. 1b, has learned the image statistics of the training set we evaluate the virtual frequency modulation $\widetilde{\mathcal{E}}$, defined as the spectrum of the generated image with the neural étendue expander and the zero-phase SLM modulation as

$$\widetilde{\mathcal{E}} = \mathcal{F}(|\mathcal{F}(\mathcal{E})|^2). \tag{5}$$

The findings in Fig. 1d confirm that the magnitude of the virtual frequency modulation $|\widetilde{\mathcal{E}}|$ resembles the magnitude spectrum of natural images within the passband. Moreover, we observe that the virtual frequency modulation pushes undesirable energy outside of the passband of the human retina as imperceptible high-frequency noise.

To further understand this property of a neural étendue expander, we consider the reconstruction loss $\mathcal{L}_T$ for a specific target image $T$. Using the zero-phase setting for the SLM as an initial point for the first-order stochastic optimization and applying Parseval's theorem places an upper bound on the reconstruction loss

$$\mathcal{L}_T = \min_{\mathcal{S}} \left\| \left( |\mathcal{F}(\mathcal{E} \odot U(\mathcal{S}))|^2 - T \right) * f \right\|_2^2 \le \frac{1}{N} \left\| \left( \widetilde{\mathcal{E}} - \mathcal{F}(T) \right) \odot \mathcal{F}(f) \right\|_2^2, \tag{6}$$

where $N$ is the pixel count of the neural étendue expander. Please see Supplementary Note 3 for further details of how this upper bound is found. Therefore, obtaining the optimal neural étendue expander, which minimizes the reconstruction loss $\mathcal{L}_T$, results in the virtual frequency modulation $\widetilde{\mathcal{E}}$ that resembles the natural-image spectrum $\mathcal{F}(T)$ averaged over diverse natural images. Also, the retinal frequency filter $\mathcal{F}(f)$ leaves the higher spectral bands outside of the human retinal resolution unconstrained. This allows the neural étendue expander to push undesirable energy towards higher frequency bands, which then manifests as imperceptible high-frequency noise to human viewers.

We investigate the image statistics preserved by the neural étendue expanders by visualizing $|\widetilde{\mathcal{E}}|$. Fig. 3e visualizes the learned expander pattern $\mathcal{E}$ for increasing étendue expansion factors, specifically for $4 \times, 16 \times, 36 \times$, and $64 \times$ expansion. Unlike uniform and binary random expanders, the learned expanders exhibit high and low frequency structures. Fig. 3f shows the corresponding virtual frequency $|\widetilde{\mathcal{E}}|$ for each étendue expansion factor. We observe that the interwoven high and low frequency patterns on each learned expander correspond to a virtual frequency that pushes noise outside of the retinal frequency bands defined by $\mathcal{F}(f)$. Furthermore, the frequency structure within the passband resembles the frequency structure of the natural image training dataset, see Fig. 3c.

To characterize the hologram reconstruction with the proposed neural étendue expander we simulate a Fourier holographic setup that has been augmented with a neural étendue expander. Fig. 3a reports qualitative examples of trichromatic and monochromatic reconstructions achieved with neural étendue expanders, binary random expanders[7], photon sieves[21], and conventional holography[1]. See Supplementary Note 2 for additional qualitative comparisons and for comparisons against uniform random expanders. The uniform random expander is constructed by assigning each pixel a phase that is uniformly randomly chosen within $[0, 2\pi]$. To ensure at least $2\pi$ phase is available for all wavelengths the $[0, 2\pi]$ phase range is defined for 660 nm. Conventional holography is subject to a low display étendue that is limited by the SLM native resolution, thus resulting in a low FOV. Photon sieves[21], binary random expanders[7], and uniform random expanders have low reconstruction fidelity, resulting in severe noise and low contrast in the generated holograms. In the case of the trichromatic holograms, both uniform and binary random expanders do not facilitate consistent étendue expansion at all wavelengths, which results in chromatic artifacts. Although the uniform random expander provides at least $2\pi$ phase coverage for all wavelengths, the variation in refractive index across wavelengths results in differing phase profiles. Thus, although the uniform random expander has the same degree of quantization as neural étendue expanders, it does not enable étendue expanded trichromatic holograms. Photon sieves scatter light equally across wavelengths but their randomized amplitude-only modulation does not allow for high-fidelity reconstruction of natural images, see Fig. 3d for quantitative assessment and Supplementary Note 2 for qualitative examples and additional metrics. Neural étendue expansion is the only technique that facilitates high-fidelity reconstructions for both trichromatic and monochromatic setups. We quantitatively verify this by evaluating the reconstruction fidelity on an unseen test dataset, where fidelity is measured in peak signal-to-noise ratio (PSNR). Fig. 3b shows that neural étendue expanders achieve over 14 dB PSNR improvement favorable to other expanders when generating $64 \times$ étendue expanded trichromatic holograms. For monochromatic holograms, neural étendue expansion achieves over 10 dB PSNR improvement. Thus, neural étendue expansion allows for an order of magnitude improvement over existing étendue expansion methods. See Fig. 3d for quantitative evaluations at different étendue expansion factors. See Supplementary Note 2 for further simulation details and more comparison examples.

We also investigate the robustness of neural étendue expansion to eye pupil movements. By initializing the learning process with a uniform random expander we bias the optimized solution towards expanders that distribute energy throughout the eyebox. As a result, the image quality of neural étendue expanded holograms is tolerant to variations in the shape and position of the viewer's eye pupil. We find that alternative methods that use quadratic phase profiles[28] concentrate the eyebox energy at fixed points and are not robust to eye pupil movements. See Supplementary Note 5 for findings. Finally, we also investigate 3D étendue expanded holograms. We find that neural étendue expansion also enables higher fidelity étendue expanded 3D color holograms. We note that existing methods on étendue expanded holography has focused on monochromatic 3D holograms[7,28,29]. Photon sieves[21] only realize 3D color holography for sparse points. See Supplementary Note 4 for a discussion of these findings.

## Discussion

In this work, we introduce neural étendue expanders as an optical element that expands the étendue of existing holographic displays without sacrificing displayed hologram fidelity. Neural étendue expanders are

learned from a natural image dataset and are jointly optimized with the SLM's wavefront modulation. Akin to a shallow neural network, this new breed of optical elements allows us to tailor the wavefront modulation element to the display of natural images and maximize display quality perceivable by the human eye. As the first learned optics for étendue expansion, we achieve étendue expansion factor 64 × with over 29 dB PSNR reconstructions, an order of magnitude improvement over existing approaches. Furthermore, neural étendue expanders support multi-wavelength illumination for color holograms. The expanders also support 3D color holography and viewer pupil movement. We envision that future holographic displays may incorporate the described optical design approach into their construction, especially for VR/AR displays. Extending our work to utilize other types of emerging optics such as metasurfaces may prove to be a promising direction for future work, as diffraction angles can be greatly enlarged by nano-scale metasurface features[37] and additional properties of light such as polarization can be modulated using meta-optics[38].

## Methods

### Simulation
We used PyTorch to design and evaluate the neural étendue expanders. See Supplementary Notes 2 and 3 for details on the optimization framework, evaluation, and analysis.

### Fabrication
The expanders are physically realized as diffractive optical elements (DOE). Fabricating the DOEs consists of several stages. The first stage consists of etching the negative of the desired pattern onto a substrate. This etching is performed with laser beam lithography. The etched substrate forms a stamp which is then pressed onto a resin mold that is mounted on a glass substrate. The resin itself contains the final pattern. The resin has a wavelength dependent refractive index that we incorporate into our design framework. For the resin we used, the refractive indices are 1.5081 for 660 nm, 1.5159 for 517 nm, and 1.5223 for 450 nm. See Supplementary Note 10 for details.

### Experimental Setup
We evaluated the neural étendue expanders using a prototype holographic display. The prototype consists of a HOLOEYE-PLUTO SLM, a 4F system, a DC block, and a camera for imaging the étendue expanded holograms. See Supplementary Notes 11 and 12 for details.

## Data availability
The code and data used to generate the findings of this study are available at https://doi.org/10.5281/zenodo.10653321.

## Code availability
The code and data used to generate the findings of this study are available at https://doi.org/10.5281/zenodo.10653321.

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

## Acknowledgements

E.T. acknowledges a Google PhD Fellowship. F.H. acknowledges an NSF CAREER Award (2047359), a Packard Foundation Fellowship, a Sloan Research Fellowship, a Sony Young Faculty Award, a Project X Innovation Award, and an Amazon Science Research Award. The authors acknowledge the use of Princeton's Imaging and Analysis Center (IAC), which is partially supported by the Princeton Center for Complex Materials (PCCM), a National Science Foundation (NSF) Materials Research Science and Engineering Center (MRSEC; DMR-2011750). The authors acknowledge Ilya Chugunov for providing natural image data. S.-H.B. acknowledges NRF Korea (RS-2023-00211658). Q.F. and W.H. acknowledge KAUST individual baseline funding. The authors acknowledge the use of the KAUST Nanofabrication Core Lab (NCL).

## Author contributions

E.T. and F.H. analyzed and designed the neural étendue expanders, performed the experiments, and led the manuscript writing. G.K., S.-H.B., A.M., P.C., Q.F. assisted in analysis and design, experiments, and writing the manuscript. N.M., F.S., W.H. assisted in writing the manuscript. D.L. and F.H. supervised the project and assisted in writing the manuscript.

## Competing interests

The authors declare no competing interests.
