## [Peer Review File · Nature Communications]

Neural Etendue Expander for Ultra-Wide-Angle High-Fidelity Holographic DisplayREVIEWER COMMENTS

Reviewer #1 (Remarks to the Author):

The authors have conducted a thorough investigation into the capabilities of conventional spatial light modulators (SLMs) for implementing a large etendue through the integration of a neural etendue expander. Although the method outlined in the manuscript is a synthesis of previously reported techniques, the authors have achieved significant advancements in the field of holographic display.

When this manuscript was initially submitted to another journal (Nature), I had several technical questions and comments. I am pleased to note that my previous concerns have been adequately addressed in the current version of the manuscript. I am in full support of publishing this manuscript in its present form. Overall, this is an excellent piece of work.

Reviewer #2 (Remarks to the Author):

This manuscript builds upon the recently published work of Kuo et al. [1], where the authors successfully demonstrated the expansion of etendue and the synthesis of wide-field images by applying random, high-frequency, phase fluctuations using an engineered phase mask in the conjugate plane of a spatial light modulator.

In this study, the authors take a similar approach but now introduce a non-random phase modulation, which is specifically optimized for the synthesis of natural images, instead of the random phase mask used in [1]. To achieve this, they took a large dataset of natural images and performed an optimization of the phase mask design to maximize the quality of rendered images globally across the entire dataset. Then, they show that the benefits of this engineered mask continue to be noticeable when the SLM is used to render holograms of previously unseen images.

The manuscript is well-written, and provides a thorough description of the experiments and results that is clear and very accessible to the reader.

Although the term "Neural etendue expander" is used to describe this method, which draws parallels to deep learning techniques, it should be noted that the device used here is a phase modulator, a conceptually identical optical element just like the one employed in [1], and does not perform nonlinear operations as typically associated with the term "Neural". The term has been used in recent publications to describe nonlinear computations, with neural holography [2] or nonlinear optical computations in diffractive optical neural networks [3].

My main reserve regarding this manuscript is that the experiments do not seem to demonstrate that the proposed method : mask optimization is what provides a significant improvement in performance compared to the previous work in [1], especially when it comes to the rendering of RGB images.

The comparison between the "neural expander" and the "random expander" is illustrated in Figure 2b, where a two-level mask is used for the random expander (Figure 5-15), while the neural expander employs an eight-level mask. The issue of using a two-level mask is that the method is particularly sensitive to a given wavelength (for which the 2-level provides a π phase shift). The fact that random etendue expansion is performed on a 2-level mask probably explains why the random mask works well for the red color but not for the green and blue parts, as observed in Figure 2D, and in [1].

If the random mask had also been generated with eight levels, similar to the "neural modulator", the etendue benefits might have extended to the other two colors as well.

Therefore, it is currently impossible to assess if the observed benefits over the prior work [1] are due to the transition from a two-level to an eight-level design of the phase mask or if they are a result of optimizing the mask for natural image rendering, as claimed by the authors. Probably, the observed improvements are a combination of both.

Therefore, for a fairer comparison, it would have been preferable for the authors to compare random and optimized etendue expanders using eight-level masks in both cases.

[1] Kuo, G., Waller, L., Ng, R. & Maimone, A. High resolution etendue expansion for holographic 249 displays. *ACM Transactions on Graphics (TOG)* 39, 1–14 (2020).

[2] Peng, Yifan, et al. "Neural holography with camera-in-the-loop training." *ACM Transactions on Graphics (TOG)* 39.6 (2020): 1-14.

[3] Lin, Xing, et al. "All-optical machine learning using diffractive deep neural networks." *Science* 361.6406 (2018): 1004-1008.

Reviewer #3 (Remarks to the Author):

Overall

Etendue expansion is the core challenge in the field of holographic displays. Holographic displays operate within the realm of light's diffraction phenomenon, and the development of high-resolution Spatial Light Modulators (SLMs) has nearly reached saturation due to practical limitations discussed in the paper. Numerous research efforts in holographic displays aim to enhance viewing parameters (FoV x eyebox, viewing angle x display size) by integrating an optical component, which is static thus does not demand any additional computation. The dynamic CGH and a static optic are jointly acquired in a single differentiable optimization pipeline. The motivation is well-received and the topic is timely. However, a critical concern arises regarding the experimental results. Despite the comprehensiveness of the analysis, as a reviewer, I find myself on the borderline of rejection. Frankly, I lean more towards rejection, considering that *Nature Communications* is a prestigious journal introducing a high-impact research with high-quality results. Given this reputation, the provided experimental results fall below the expected standard.

Strong parts

- Joint Optimization Pipeline: The authors propose a joint optimization pipeline for CGH and etendue expander, which involves a 3-bit phase mask with a DOE. This optimization is implemented with a natural image dataset and demonstrates competence compared to

previous methods. Notably, the etendue expander learns the attributes of the natural image, resulting in superior performance compared to other approaches in simulated images.

- Comprehensive Comparison: The proposed approach is extensively compared both in simulation and experiment. In simulation, it is benchmarked against different scenarios, including photo sieves and uniform binary random expanders, as shown in Table S1. In experimental settings, comparisons are made with baseline conditions (without expansion) and binary random expanders, as presented in Figure S1-S6.

- Thorough Analysis: The authors perform a comprehensive analysis covering various aspects, including the expander's performance, SLM resolution, and eyebox. These analyses are technically sound and contribute to a robust evaluation of the proposed approach.

Minor comments

- Some figures could potentially mislead readers. For instance, in Figure 1a and b, the placement of the neural etendue expander may give the impression that it simultaneously enlarges both the FoV and the eyebox with an identical eyepiece lens. However, if the field is reconstructed at the focal length of the lens, the eyebox (located at the back focal point of the lens) varies depending on the total diffraction angle, while the FoV remains constant. This potential confusion can also be extended to Figure 2a. If the hologram is generated as a Fourier-type, it requires an additional lens in front of the camera to effectively demonstrate Etendue expansion in the FoV dimension. Therefore, Figure 2a should be aligned with Figure S20 to enhance comprehension.

- In lines 32-33, it's important to note that in table-top displays, the commonly used metric is the viewing angle \times display size. This term corresponds to the eyebox \times FoV in near-eye display configurations.

Limitation

- Despite the comprehensiveness and thoroughness of the analysis, some critical issues in the experimental section should be addressed. It is indeed surprising that all the presented results are based on temporal multiplexing of 20 random phase holograms, tone-mapped (although the authors claim that these were equally performed across all cases), and the images are generated by subtracting the expander's DC term from the captured image. It is

essential to provide the raw images, showing a single frame without tone-mapping and without subtracting the DC terms from the captured image.

- There are methods available to bridge the gap between simulated and displayed scenes, as demonstrated by previous studies such as Peng et al. (2020) and Chakravarthula et al. (2020), and more advanced approaches have been introduced recently by Choi et al. (2021) and Jang et al. (2022). As the CGHs are Fourier-type thus, generating randomized field, the system necessitates precise model training procedures. It is not mandatory to include these calibration procedures in every case, but the research should elucidate how the display can achieve optimal performance without relying on 'manual' post-processing techniques.

Discussion

- The work used a 8-bit 1K phase-only SLM with a frame rate of 60 Hz. This choice of a lower-resolution SLM model minimizes practical artifacts such as phase fluctuations and crosstalk between pixels. These issues tend to be more prevalent in higher-resolution (8K) SLM models with faster frame rates. Therefore, a more in-depth discussion is required to explore ways to enhance the robustness of the joint design.

- The research utilized a 3-bit phase mask, and the level of quantization is closely linked to aspects like error diffusion and the cost of DOE manufacturing. A more comprehensive discussion on this aspect could provide valuable insights into the study's findings and implications.

Point-by-point Response: “Neural Étendue Expander for Ultra-Wide-Angle High-Fidelity Holographic Display.”

Reviewer #1 Comment: When this manuscript was initially submitted to another journal (Nature), I had several technical questions and comments. I am pleased to note that my previous concerns have been adequately addressed in the current version of the manuscript. I am in full support of publishing this manuscript in its present form. Overall, this is an excellent piece of work.

Response: We thank the reviewer for their detailed feedback for the previous submission to Nature which has helped us improve our manuscript. We are pleased that they found that this work represents a substantial contribution to the field.

Reviewer #2 Comment: My main reserve regarding this manuscript is that the experiments do not seem to demonstrate that the proposed method: mask optimization is what provides a significant improvement in performance compared to the previous work in [1], especially when it comes to the rendering of RGB images.

The comparison between the "neural expander" and the "random expander" is illustrated in Figure 2b, where a two-level mask is used for the random expander (Figure 5-15), while the neural expander employs an eight-level mask. The issue of using a two-level mask is that the method is particularly sensitive to a given wavelength (for which the 2-level provides a pi phase shift). The fact that random étendue expansion is performed on a 2-level mask probably explains why the random mask works well for the red color but not for the green and blue parts, as observed in Figure 2D, and in [1].

Therefore, it is currently impossible to assess if the observed benefits over the prior work [1] are due to the transition from a two-level to an eight-level design of the phase mask or if they are a result of optimizing the mask for natural image rendering, as claimed by the authors. Probably, the observed improvements are a combination of both.

Therefore, for a fairer comparison, it would have been preferable for the authors to compare random and optimized etendue expanders using eight-level masks in both cases.

Response: We thank the reviewer for their detailed feedback. However, we note that this feedback is the same as what we received for the first version of our manuscript that we had submitted previously to Nature. In the revised version of the manuscript that we have submitted to Nature Communications we have added extensive comparisons that clearly validate the advantages of neural étendue expanders over 8-level random expanders in all cases. These new additional experiments indeed address the very concerns you raise in your comment. See the new additional Figure 3b and 3d for reconstruction quality comparisons against 8-level expanders for both monochromatic and trichromatic reconstruction and see Supplementary Figures S10, S11, S12, S13 for the corresponding qualitative comparisons. See Supplementary Note 4 for comparisons against the 8-level random expander for 3D color hologram reconstructions where we demonstrate over 5 dB PSNR improvement. All revisions have been marked with blue color.

We emphasize that over 10 dB PSNR improvement over random expanders (binary and uniform) in reconstruction quality holds even in the case of monochromatic holograms,

see Figure 3b. We emphasize the monochromatic case because random expanders produce the same reconstruction quality regardless of quantization level in this case. This is because the spectral response of the random expander does not change appreciably with varying quantization level whereas the neural étendue expander learns a spectral response that assists with reconstruction of natural images, see Figure 3c. Thus, we emphasize that the core contribution of our method is in being able to cater the spectrum of the expander to natural images in both the monochromatic and trichromatic cases.

Furthermore, we note that it is the biasing of the spectrum towards natural images that primarily contributes to the improvement in reconstruction quality, not the differences in quantization levels. We theoretically demonstrate this with the upper bound on reconstruction error shown in Eq. (6). This upper bound is tightest when the spectrum of the expander is equal to the average spectrum of the training set. We observe that the expander indeed learns this spectrum as shown in Figure 3c and Supplementary Figure S14. Furthermore, we have also performed experiments that show higher quality monochromatic and trichromatic reconstructions across any number of height levels. See Supplementary Note S8 where we demonstrate that neural étendue expanders are superior to random expanders for two, four, and eight height levels.

Additionally, we have also provided Supplementary Code that allows for reproduction of the comparisons against the 8-level random expanders.

As a side note, we find that all three reviewers of our initial submission to Nature raised the same concern regarding 8-level random expanders, which we have addressed as discussed above. We note that other concerns raised for the initial submission to Nature have also been addressed in this submission to Nature Communications. These include experiments that demonstrate the robustness of neural étendue expanded holograms to eye pupil movements, see Supplementary Note 5, and experiments that showcase 3D étendue expanded color holograms, see Supplementary Note 4.

Reviewer #2 Comment: Although the term "Neural étendue expander" is used to describe this method, which draws parallels to deep learning techniques, it should be noted that the device used here is a phase modulator, a conceptually identical optical element just like the one employed in [1], and does not perform nonlinear operations as typically associated with the term "Neural". The term has been used in recent publications to describe nonlinear computations, with neural holography [2] or nonlinear optical computations in diffractive optical neural networks [3].

Response: We agree with the reviewer that the system is not performing non-linear operations. We chose the term "neural" to emphasize the biasing of our expander towards natural image reconstruction and how we achieved this bias through stochastic gradient descent training on a natural image dataset, which we is akin to the training of neural networks in the deep learning age.

Reviewer #3 Comment: Despite the comprehensiveness and thoroughness of the analysis, some critical issues in the experimental section should be addressed. It is indeed surprising that all the presented results are based on temporal multiplexing of 20 random phase holograms, tone-mapped (although the authors claim that these were equally performed across all cases), and the images are generated by subtracting the expander's DC term from the captured image. It is

essential to provide the raw images, showing a single frame without tone-mapping and without subtracting the DC terms from the captured image.

Response: We have included revisions that addresses these experimental concerns. Individual raw captures without any modification are shown in Supplementary Figures S7, S8, and S9. We constructed our hardware prototype in the exact same manner as in Kuo et al. (2020). Furthermore, we confirmed the correctness of our hardware prototype by comparing the single frame raw images shown in S7, S8, and S9 for the binary random expander to those shown in Figure 6 of Kuo et al. (2020). We observe improvement in quality of the holograms produced by neural étendue expanders over random expanders even within the raw unmodified images.

Despite our best efforts to manufacture high quality expander elements we observed that the diffraction efficiency of the DOEs was not 100%. This results in some amount of DC leakage which obscured our observation of the modulated component of the DOE. Thus, we minimally suppressed the DC component in order to better visualize the hologram. As stated in Supplementary Note 2: Additional Experimental Results, this DC term could be eliminated in the future through a tilted off-axis construction of the display or through hardware-in-the-loop methods that account for the DC term. We have revised the text in Supplementary Note 2 to clarify these points.

Secondly, we clarify that the only tone-mapping performed is white balancing of the different color channels which is equivalent to changing the laser power of the laser source. We did this because we captured all raw data at the same camera exposure level and laser power settings to ensure that the raw data does not contain saturated pixels. While this approach avoids clipped pixels, the white balance of a captured color hologram with respect to the target image could be incorrect. Thus, we correct the white balance by multiplying each color channel by a scalar scale factor and this is shown explicitly in our Supplementary Code. We have revised the text in Supplementary Note 2 to clarify these points.

Thirdly, we emphasize that temporal multiplexing is not a necessary component of our proposed method. Indeed, the training of the neural étendue expander was performed without any temporal multiplexing, we have clarified this in the main manuscript and in a new Supplementary note (Supplementary Note 7: Temporal Analysis). We have also included reconstruction scores for varying temporal multiplexing quantities ranging from a single frame to 20 frames in Supplementary Table S4. The table shows that neural étendue expanders always achieve over 10 dB PSNR improvement in reconstruction quality regardless of the number of temporal frames used. Furthermore, the reconstruction quality does not change appreciably after 3 frames. These results can be reproduced with the provided code. We also note that the Supplementary Video shows experimental results that only uses 3 frames for time averaging in order to maintain a 20 Hz framerate, as described in Supplementary Video 1: Experimental Assessment for Dynamic Scenes.

Lastly, we re-emphasize that any modifications are applied equally to the holograms produced by both random expanders and neural étendue expanders for fair comparison. This can also be seen in the Supplementary Code that we have included with our submission in the “Experimental” folder.

Reviewer #3 Comment: There are methods available to bridge the gap between simulated and displayed scenes, as demonstrated by previous studies such as Peng et al. (2020) and Chakravarthula et al. (2020), and more advanced approaches have been introduced recently by Choi et al. (2021) and Jang et al. (2022). As the CGHs are Fourier-type thus, generating randomized field, the system necessitates precise model training procedures. It is not mandatory to include these calibration procedures in every case, but the research should elucidate how the display can achieve optimal performance without relying on 'manual' post-processing techniques.

Response: We recognize the progress that has been made in bridging simulation and experimental reconstruction within the holography community. However, these works have primarily demonstrated calibration schemes for Fresnel-type holograms with smooth phase (field). As the reviewer notes, Fourier-type holograms exhibit fully randomized fields and it is unclear whether existing calibration approaches are applicable.

Nevertheless, we have included additional discussion in Supplementary Note 2: Additional Experimental Results on possible directions for closing the gap between simulation and experiment. Non-idealities in the SLM could be calibrated for by observing its ability to generate a focused dot of light. Experimentally capturing the focused dot and iteratively improving the maximum intensity of the dot through a hardware-in-the-loop approach could offer an approach for calibrating out non-idealities of the SLM. Non-idealities of the DOE, specifically the non-ideal diffraction efficiency, could be compensated for by experimentally capturing its response and incorporating it into the CGH method.

As a side note, the randomized fields of our étendue expanded Fourier holograms confers advantages in robustness to pupil movement. Supplementary Note 5: Pupil and Eyebox Analysis shows that holograms generated with our learned expanders are robust to eye pupil movement whereas alternative approaches that use a simple lens to expand the FOV while shrinking the eyebox do not have this robustness.

Reviewer #3 Comment: The work used a 8-bit 1K phase-only SLM with a frame rate of 60 Hz. This choice of a lower-resolution SLM model minimizes practical artifacts such as phase fluctuations and crosstalk between pixels. These issues tend to be more prevalent in higher-resolution (8K) SLM models with faster frame rates. Therefore, a more in-depth discussion is required to explore ways to enhance the robustness of the joint design.

Response: We agree with the reviewer that lower-resolution SLMs would exhibit less non-ideal behavior than higher-resolution SLMs. We have included additional discussion in Supplementary Note 6: Resolution Analysis on this topic. Specifically, modeling the field fringing response of high-resolution SLMs and incorporating that response into the training of the expander could be a path towards enhancing robustness in the future. Indeed, recent works by Markley et al. (2022), Kuo et al. (2023), and Nam et al. (2023) have shown that field fringing can be modeled as a convolution of the SLM phase with a small learnable kernel.

Reviewer #3 Comment: The research utilized a 3-bit phase mask, and the level of quantization is closely linked to aspects like error diffusion and the cost of DOE manufacturing. A more comprehensive discussion on this aspect could provide valuable insights into the study's findings and implications.

Response: We have provided an additional discussion in Supplementary Note 9: Expander Fabrication on the connection between quantization and DOE manufacturing. We produced our DOEs through resin stamping. As such, the largest contributor to cost comes from the manufacturing of the mold. Stamping additional DOEs after the mold is made is cheap. This differs from alternative processes such as gas-etching where the write times and cost scales with the number of quantization levels. As such, we believe that the manufacture of the expanding element should not introduce a significant cost.

Furthermore, we have included a new Supplementary Note (Supplementary Note 8: Quantization Analysis) where we evaluate the effect of quantization on reconstruction quality. We observe that neural étendue expanders facilitate higher reconstruction quality even with heavy quantization, see Supplementary Table S5.

Reviewer #3 Comment: Some figures could potentially mislead readers. For instance, in Figure 1a and b, the placement of the neural étendue expander may give the impression that it simultaneously enlarges both the FoV and the eyebox with an identical eyepiece lens. However, if the field is reconstructed at the focal length of the lens, the eyebox (located at the back focal point of the lens) varies depending on the total diffraction angle, while the FoV remains constant. This potential confusion can also be extended to Figure 2a. If the hologram is generated as a Fourier-type, it requires an additional lens in front of the camera to effectively demonstrate Étendue expansion in the FoV dimension. Therefore, Figure 2a should be aligned with Figure S20 to enhance comprehension.

Response: We agree with these comments and have improved our manuscript accordingly. Specifically, we have incorporated an eyepiece illustration into Figure 1a and b to clarify that the holograms in this work are Fourier-type. We have also adjusted the labeling of the optical elements in Figures 1a, 1b, and 2a to match the labeling used in Supplementary Figure S23 (previously Supplementary Figure S20). Figures 1a, 1b, and 2a now have explicit labels for the Fourier transforming lens and the eyepiece/imaging lens. We have also revised all of the figure captions to clarify that the holograms are all Fourier-type and that the configuration uses a Fourier transforming lens and an eyepiece/imaging lens.

Reviewer #3 Comment: In lines 32-33, it's important to note that in table-top displays, the commonly used metric is the viewing angle x display size. This term corresponds to the eyebox x FoV in near-eye display configurations.

Response: We thank the reviewer for this editorial suggestion. We have incorporated the distinction between metrics used for table-top displays versus near-eye displays into the revised manuscript.

REVIEWER COMMENTS

Reviewer #1 (Remarks to the Author):

The revised manuscript has greatly improved in terms of clarity. I fully support its publication.

Reviewer #2 (Remarks to the Author):

I would like to send my apology to the authors for failing to notice that some of my comments had been addressed in a revised version submitted to a new journal. I was mostly surprised to see a resubmission so rapidly submitted and assumed that the authors had simply resubmitted the manuscript in another journal without addressing comments, which is, unfortunately, a common practice in science, and one of the heartbreaking experiences of being a reviewer. This is very much not the case here, and I realize that I should have done a better re-reading of the manuscript.

I had this opportunity, and I am pleased to report that the authors have adequately addressed some of my comments, and some important issue raised by the other reviewers. In light of these updates, I believe that the manuscript is suitable for publication.

Reviewer #3 (Remarks to the Author):

I highly appreciate the authors' effort in addressing the raised concerns (DOE quantization, temporal analysis, etc.), especially in providing numerous results in the validation section. I would be glad to change my first review, and I think this work can be published in Nature Communications after a minor revision.

Upon examining the raw images provided during the revision period, I noticed significant improvements in single-frame single-color (central wavelength of 660 nm) results with a high (64x) etendue expansion ratio. Furthermore, when comparing the first row of Figure S9 (x16) and the first row of Figure S7 (x64), the improvement becomes more noticeable in the latter case. This improvement, depending on the expansion ratio observed in the

experiment, emphasizes the contribution of the joint design of the expander and CGH.

Since Fourier-type holograms reconstruct the complex field with randomized phase, I understand that it is very challenging to validate and compare them with the previous state-of-the-art work [Kuo et al. 2020] in experimental settings. In simulations, optimizations are conducted based on the ideal model, and the difference in the simulated/estimated metrics (PSNR) decreases in the actual implementation due to the discrepancy between the ideal and actual models [Peng et al. 2020]. There are many aspects that make the actual model not ideal (aberration, distortion, SLM non-linearity, etc.), and many aspects are discussed in the previous publications. In this perspective, I would like to see an analysis of the model mismatch problem, which can be more prevalent in high-resolution SLMs with a small pixel pitch. For instance, the pixels may present different phase values with errors following a Gaussian distribution with a certain standard deviation, or the fringing field effect can be present, as discussed in the response letter. This can highlight why phase fluctuation is a critical issue in holographic displays, especially when generating a randomized field, as in this work.

Although the work presents proof-of-concept results, it is not advisable to over-emphasize the use of an 8K SLM with a 64x etendue expansion ratio throughout the entire paper. Validations are performed with an 8-bit 1K SLM, and the results are provided with temporal multiplexing of 20 frames in the main paper. To the best of my knowledge, there are no commercially available 8-bit 1K SLMs that offer a frame rate of 60 (flicker threshold) x 20 (temporal multiplexing) x 3 (color) Hz. Thus, this TM number should be stated in Figure 2. However, these limitations of high-resolution active devices highlight the advantages of combining low-resolution active devices with nano-structured passive devices.

Point-by-point Response: “Neural Étendue Expander for Ultra-Wide-Angle High-Fidelity Holographic Display.”

Reviewer #1 Comment: The revised manuscript has greatly improved in terms of clarity. I fully support its publication.

Response: We thank the reviewer for their detailed feedback for the previous versions which has helped us improve our manuscript. We are pleased that they support the publication of our manuscript.

Reviewer #2 Comment: I had this opportunity, and I am pleased to report that the authors have adequately addressed some of my comments, and some important issue raised by the other reviewers. In light of these updates, I believe that the manuscript is suitable for publication.

Response: We thank the reviewer for their detailed feedback for the previous versions which has helped us improve our manuscript. We are pleased that they support the publication of our manuscript.

Reviewer #3 Comment: I highly appreciate the authors' effort in addressing the raised concerns (DOE quantization, temporal analysis, etc.), especially in providing numerous results in the validation section. I would be glad to change my first review, and I think this work can be published in Nature Communications after a minor revision.

Upon examining the raw images provided during the revision period, I noticed significant improvements in single-frame single-color (central wavelength of 660 nm) results with a high (64x) étendue expansion ratio. Furthermore, when comparing the first row of Figure S9 (x16) and the first row of Figure S7 (x64), the improvement becomes more noticeable in the latter case. This improvement, depending on the expansion ratio observed in the experiment, emphasizes the contribution of the joint design of the expander and CGH.

Since Fourier-type holograms reconstruct the complex field with randomized phase, I understand that it is very challenging to validate and compare them with the previous state-of-the-art work [Kuo et al. 2020] in experimental settings. In simulations, optimizations are conducted based on the ideal model, and the difference in the simulated/estimated metrics (PSNR) decreases in the actual implementation due to the discrepancy between the ideal and actual models [Peng et al. 2020]. There are many aspects that make the actual model not ideal (aberration, distortion, SLM non-linearity, etc.), and many aspects are discussed in the previous publications. In this perspective, I would like to see an analysis of the model mismatch problem, which can be more prevalent in high-resolution SLMs with a small pixel pitch. For instance, the pixels may present different phase values with errors following a Gaussian distribution with a certain standard deviation, or the fringing field effect can be present, as discussed in the response letter. This can highlight why phase fluctuation is a critical issue in holographic displays, especially when generating a randomized field, as in this work.

Response: We thank the reviewer for their detailed feedback for the previous versions which has helped us improve our manuscript. We have included an additional Supplementary Note that analyzes the impact of SLM phase non-idealities such as phase

fluctuations and field fringing. Qualitative and quantitative results for this analysis can be found in Supplementary Note 10: SLM Phase Non-Ideality Analysis. For phase fluctuation analysis we apply per-pixel Gaussian noise to the SLM phase pattern as suggested by the reviewer. For field fringing analysis, we upsample the SLM pattern and then we apply a Gaussian kernel to the SLM pattern to model the field fringing effect. We vary the standard deviation of the Gaussian noise and the radius of the field fringing kernel to analyze the impact of increasing SLM phase non-ideality. We also analyze the impact of simultaneously applying both non-idealities to the SLM. We validate that neural étendue expanders improve hologram reconstruction quality even in the presence of SLM phase non-ideality.

Reviewer #3 Comment: Although the work presents proof-of-concept results, it is not advisable to over-emphasize the use of an 8K SLM with a 64x étendue expansion ratio throughout the entire paper. Validations are performed with an 8-bit 1K SLM, and the results are provided with temporal multiplexing of 20 frames in the main paper. To the best of my knowledge, there are no commercially available 8-bit 1K SLMs that offer a frame rate of 60 (flicker threshold) x 20 (temporal multiplexing) x 3 (color) Hz. Thus, this TM number should be stated in Figure 2. However, these limitations of high-resolution active devices highlight the advantages of combining low-resolution active devices with nano-structured passive devices.

Response: We thank the reviewer for their editorial feedback. We have removed most mentions of the 8K SLM from the main manuscript. We have kept only one mention of the 8K SLM in the main manuscript's analysis section to point the reader to the resolution analysis section in the Supplementary Information. We have also revised the caption of Figure 2 to mention the number of frames used for temporal multiplexing.

REVIEWERS' COMMENTS

Reviewer #3 (Remarks to the Author):

I thank the authors for increasing clarity and providing sufficient detail. I would also like to thank the other reviewers for their contributions to this excellent work. In this updated version, I support the publication of this work.